# REIV-TOXO Project: Results from a Spanish cohort of congenital toxoplasmosis (2015–2022). The beneficial effects of prenatal treatment on clinical outcomes of infected newborns

Borja Guarch-Ibáñez[1,2,3☯] *, Clara Carreras-Abad[3,4☯], Marie Antoinette Frick[3,5,6], Daniel Blázquez-Gamero[3,7], Fernando Baquero-Artigao[3,8,9,10], Isabel Fuentes[11], the Spanish Research Network of Congenital Toxoplasmosis (REIV-TOXO) group[¶], Pere Soler-Palacin[5,6] *

1 Pediatric Infectious Diseases Unit, Pediatrics Department, Hospital Universitari de Girona Dr. Josep Trueta, Girona, Catalonia, Spain, 2 Universitat de Girona (UDG), Girona, Catalonia, Spain, 3 Congenital Infections Working Group, Spanish Society of Pediatric Infectious Diseases (SEIP), Spain, 4 Pediatric Infectious Diseases Unit, Pediatrics Department, Hospital Universitari Germans Trias i Pujol, Badalona, Catalonia, Spain, 5 Pediatric Infectious Diseases and Immunodeficiencies Unit, Hospital Infantil Vall d'Hebron, Barcelona, Catalonia, Spain, 6 Vall d'Hebron Research Institute, Barcelona, Catalonia, Spain, 7 Pediatric Infectious Diseases Unit, Hospital Universitario 12 de Octubre, Madrid, Spain, 8 Pediatric Infectious Diseases Unit, Hospital Universitario La Paz, Madrid, Spain, 9 Universidad Autónoma de Madrid, Madrid, Spain, 10 CIBERINFEC, Instituto de Salud Carlos III, Madrid, Spain, 11 Toxoplasmosis and Intestinal Protozoa Unit, Centro Nacional de Microbiología, Instituto de Salud Carlos III, Madrid, Spain

☯ These authors contributed equally to this work.
¶ Membership of the Spanish Research Network of Congenital Toxoplasmosis (REIV-TOXO) is provided in Supporting Information file S1 Acknowledgments
* bguarch.girona.ics@gencat.cat (BGI); pere.soler@vallhebron.cat (PSP)

**Data Availability Statement:** There are ethical restrictions on sharing data set because it contains

## Abstract

### Background

Some regions of Spain are withdrawing their pregnancy screening program for congenital toxoplasmosis (CT). The Spanish Research Network of Congenital Toxoplasmosis (REIV-TOXO) was created to describe the current status of CT in Spain. The aims of this study were to describe the epidemiological and clinical characteristics of CT and to evaluate the effect of prenatal treatment on clinical outcomes to inform decision-making policies.

### Methods

Ambispective observational study including CT cases recorded in the REIV-TOXO database that includes 122 hospitals (2015–2022). Inclusion criteria were one or more of the following: positive PCR in maternal amniotic fluid; positive *Toxoplasma gondii*-specific IgM or IgA antibodies at birth; positive PCR in the placenta, newborn blood, urine or CSF; increase of specific IgG levels during infant follow-up; or specific IgG persistence beyond age 12 months.

potentially identifying and sensitive patient information and this restriction is imposed by the Research Ethics Committee. To request data please contact the Research Ethics Committee of Hospital Universitari de Girona Dr. Josep Trueta (ceic. girona.ics@gencat.cat). There is no third party organization that has access to the complete data included in the article.

**Funding:** This study was partially funded by the national project FIS AESI PI21CIII/00031- Health Research Fund, Carlos III Health Institute, Ministry of Science and Innovation - (ICIII to BGI and IFC) and by a private donation from the Bescos-Manau family (BMF to BGI and PSP) to promote research in congenital toxoplasmosis. The funders had no role in study design, data collection and analysis, decision to publish, or preparation of the manuscript.

**Competing interests:** The authors have declared that no competing interests exist.

## Findings

Fifty-six newborns (54 pregnancies) were included. Prenatal screening allowed 92.8% of cases to be identified. The time of maternal infection was well documented in 90.7% of cases, with 61.1% occurring in the third trimester. A total of 66.6% (36/54) pregnant women received antiparasitic treatment: 24/36 spiramycin, 8/36 pyrimethamine, sulfadiazine, and folinic acid, and 4/36 both treatments sequentially. Most cases were asymptomatic at birth (62.5%, 35/56), and 84% (47/56) newborns completed one year of treatment. Median follow-up was 24 months (IQR = 3–72): 14.2% children exhibited new complications, mainly ocular. Newborns born to mothers treated prenatally had four-fold lower risk of CT clinical features at birth (p = 0.03) and six-fold lower risk of further complications during follow-up (p = 0.04) with no treatment-related differences during pregnancy.

## Conclusions

While diagnosis based only on neonatal assessment misses a significant number of CT cases, prenatal screening allows treatment to be started during pregnancy, with better clinical outcomes at birth and during follow-up. REIV-TOXO provides valuable information about CT in Spain, highlighting the need for universal maternal screening.

## Author summary

Spain is presumed to have a low burden of congenital toxoplasmosis, but the true incidence of the disease is currently unknown. To describe and analyze the status of congenital toxoplasmosis in Spain, in 2018 we created a multicenter registry known as the Spanish Research Network of Congenital Toxoplasmosis (REIV-TOXO) to collect data on all children diagnosed with congenital toxoplasmosis born in Spain from January 1, 2015 to the present. In this paper, we describe the main characteristics of cases included in REIV-TOXO between 2015 and 2022. We also focused on the effect of prenatal treatment on clinical outcomes at birth and during follow-up, since its effectiveness has been questioned in the past and may raise several concerns. One of these issues is the actual benefit of serological screening during pregnancy as part of a public health strategy to mitigate the disease. Although the implementation of prenatal screening is still under debate, it remains the only tool available that allows any potentially infected children to be identified and afford them an opportunity for early follow-up and treatment. In view of the progressive withdrawal of prenatal screening in Spain in recent years, the results of our cohort are relevant, as they provide data that could aid public health decision-making policies.

## Introduction

Toxoplasmosis is a major zoonotic illness worldwide, affecting about 25% to 30% of the population [1,2]. Infection is caused by the intracellular parasite *Toxoplasma gondii* and can be congenital or acquired, with the latent form persisting throughout the individual's lifetime. Congenital toxoplasmosis (CT) results from transplacental transmission in pregnant women who acquire primary infection or experience reactivation during pregnancy [1,3,4].

Although the infection is usually asymptomatic in the mother, the infected newborn may exhibit a broad spectrum of clinical manifestations, ranging from asymptomatic forms to severe and irreversible defects with ocular, neurological, and/or systemic involvement, miscarriages, and stillbirths [5]. Asymptomatic forms at birth are more common, but children with the condition may experience complications caused by reactivation of latent infection at any time in their life, primarily as retinochoroiditis, which can cause vision loss and even blindness [3,5–8].

Although CT is present on a global scale, there are significant geographic differences in the incidence and severity of the illness [3,5]. In Europe, its impact is lower than in areas of Latin America, Central Africa, and the Middle East due to differences in socioeconomic situation and hygiene-dietary habits, parasite-related factors (low diversity and predominance of the type II genotype), host factors, and the capacity of the health system in each country to handle early screening and treatment of all cases [1,5,9–16]. However, European countries show a high variability in the number of cases recorded due to a lack of reporting, limitations in the disease surveillance programs, and progressive withdrawal of prenatal screening in different areas [17].

At present, prenatal serological screening is the only diagnostic tool available to identify all potential children that could be affected by CT and provide them with adequate diagnosis, treatment, and follow-up. However, there is no international consensus on screening for this illness in pregnant women. Whereas some European countries such as France, Slovenia, Austria, and Italy, promote regular serological screening, others have discontinued these programs, as was the case for prenatal screening in Germany, Switzerland, and Denmark [18,19]. The reasons for these differences are multiple and debatable. These include the cost-benefit of screening programs, the lack of randomized controlled studies providing evidence for early diagnosis and treatment, and the possible anxiety produced by an early detection [10,20,21]. Consequently, each country has implemented different strategies to prevent CT, according to the prevalence of the disease, pathogen virulence, and health care policies in their respective regions [19–24].

In France, a country with a well-established prenatal screening system and a maintained decline in seroprevalence among pregnant women (31,6% in 2016), 200 cases of CT are reported each year, with an annual incidence of 2 per 10,000 live births [19]. A similar decrease in toxoplasmosis seroprevalence among pregnant women has been identified in Spain, with an overall prevalence of 24.4% over the last three decades (95% CI: 21.2–28.0%) [25]. Despite the differences, the estimated incidence rate of CT in Spain and France in 2013 was similar [11]. Subsequently, a retrospective study showed fewer hospitalizations in Spain due to CT between 2010 and 2018, with a hospitalization rate of 0.61 per 10,000 live births in 2018 [26]. On the other hand, only 7 cases of CT were reported in Spain to the European Centre for Disease Prevention and Control (ECDC) for the 2017–2021 period [17]. Despite CT being a reportable disease, it is presumed to be underreported in the official Spanish registries. Moreover, prenatal screening is still under discussion in Spain, with implementation varying widely and testing either not performed or performed only once per trimester [27,28]. Consequently, the epidemiology and disease burden of CT in Spain remain unknown. This led to the creation of the Spanish Research Network of Congenital Toxoplasmosis (REIV-TOXO) in 2018 to collect information on the epidemiological, clinical, diagnostic, therapeutic, and follow-up characteristics of newborns infected with *T. gondii* in Spain.

The aim of the present study is to characterize the situation of CT in Spain from 2015 to 2022 based on all CT cases included in the REIV-TOXO database and to evaluate the effect of prenatal treatment on clinical outcomes at birth and during follow-up, thus providing data that could aid in public health decision-making policies.

## Methods

### Ethics statement

The project was approved by the Research Ethics Committee of Girona for the Hospital Universitari de Girona Doctor Josep Trueta (CEIm code 2018.027) as the coordinating center and by all other participating centers, and received scientific endorsement from the Spanish Society of Pediatric Infectious Diseases (SEIP) and the Spanish Pediatrics Association (AEP). All parents gave written informed consent for the pseudonymized case data to be included in the REIV-TOXO database.

An ambispective observational multicenter study was conducted of children with CT recorded in the national REIV-TOXO database between 1 January 2015 and 30 June 2022. REIV-TOXO includes 122 hospitals in Spain and conducts active research of cases to collect different variables on children born in Spain and diagnosed with CT using the Research Electronic Data Capture platform (REDCap; 7.6.5, Vanderbilt University, Nashville, TN, USA).

A CT case was defined as meeting one or more of the following criteria: positive polymerase chain reaction (PCR) in amniotic fluid (AF); presence of *T. gondii*-specific immunoglobulins (Ig) (IgM or IgA); positive PCR for *T. gondii* in the newborn's blood, urine, cerebrospinal fluid (CSF) or in the placenta; increase in specific IgG levels during the infant's follow-up; or persistence of specific IgG beyond 12 months of life. Last, children who met the criteria for CT were recorded in the REIV-TOXO registry.

Prenatal treatment with spiramycin (SPI) and/or pyrimethamine, sulfadiazine, and folinic acid (PSA) was defined as receiving one or both treatments during pregnancy for at least 28 days. Treatment of either antiparasitic drugs received for less than 7 days was considered to be "no prenatal treatment." Pregnant women treated between 7 and 27 days were considered to have "uncertain prenatal treatment" because the effect of the short course of treatment was unknown.

Serology techniques used to detect *T. gondii*-specific IgA, IgM, and IgG antibodies varied according to the internal protocol at each facility. Molecular diagnosis was performed by using PCR to detect *T. gondii* DNA in the samples analyzed (AF, placenta, newborn's blood, urine, and/or CSF), with the ones used most often being nested-PCR to amplify a B1 sequence and real-time PCR to amplify the rep 529 fragment of the parasite [29,30].

Illness at birth was classified as asymptomatic or symptomatic CT; the clinical presentations are listed in Table 1. The presence of abnormal blood or CSF results was not included as an inclusion criterion for symptomatic CT cases.

**Table 1. Clinical forms of CT at birth.**

| |
|---|
| - **Symptomatic CT**: Presence of consistent signs or symptoms on physical and ophthalmological examination as well as imaging findings. |
| • **Ocular CT:** Findings on ophthalmological examination consistent with retinochoroiditis, cataracts, acute retinal necrosis, retinal dysplasia, microphthalmia, and/or optic atrophy. |
| • **CT with neurological involvement:** Described as seizures, microcephaly (HC < –2SD*), macrocephaly (HC > +2SD*) and/or the following imaging findings on transfontanelle ultrasound and/or brain MRI: hydrocephalus, intracranial calcifications, and/or cerebral parenchyma lesions. |
| • **CT with generalized disease:** Presence of jaundice, enlarged spleen, enlarged lymph nodes, hepatomegaly, exanthema, and/or pneumonitis. |
| - **Asymptomatic CT**: No signs or symptoms consistent with symptomatic CT. |

*According to the World Health Organization growth standards [31] or the standards taken from Carrascosa et al. [32]

CT = congenital toxoplasmosis; HC = head circumference; MRI = magnetic resonance; SD = standard deviation.

Adverse drug reactions were recorded for both the mother and the newborn: presence of hypersensitivity reactions (Stevens–Johnson syndrome or Lyell syndrome), hives, liver toxicity (defined as elevated aspartate aminotransferase/alanine aminotransferase $\geq$ two-fold the upper limit of normal), neutropenia (neutrophils below 1000/mm$^3$), aplastic anemia, thrombocytosis (platelets above 450,000/mm$^3$), gastrointestinal symptoms, crystalluria, and nephrotoxicity (defined as urea levels above 28 mg/dL and/or creatinine above 1.06 mg/dL).

For the statistical analysis, qualitative variables are expressed as absolute and relative frequencies, whereas quantitative variables are expressed as the mean and standard deviation or the median and interquartile range (IQR), depending on the distribution of the variable.

A logistic regression model of the variables of interest was performed to analyze the effect of treatment during pregnancy, adjusting for whether or not the mother had received treatment during pregnancy with SPI and/or PSA. Cases included in the "uncertain prenatal treatment group" were excluded from this analysis. The same model was also used to compare the two regimens received: SPI versus PSA. A multivariate logistic regression model, adjusted for the type of treatment in the pregnant mother and for the time of maternal infection, was performed to control for potential effects of the latter. The effect of any statistically significant variables is expressed as the odds ratio (OR) and the respective 95% confidence interval (CI). The significance level was set at 95% ($p < 0.05$) for all analyses, and all analyses were performed using SAS v9.4 (SAS Institute Inc., Cary, NC, USA).

## Results

Between 2015 and 2022, a total of 72 cases of CT were identified by an active search for cases; 56 cases from 54 pregnancies (52 singleton and 2 twin pregnancies) were finally included in the REIV-TOXO database.

Catalonia (21 cases) was the autonomous community with the highest number of cases included, followed by the Community of Madrid and the Valencian Community, with 12 and 9 cases, respectively. Trends over time for patient screening during the study period are depicted in Fig 1. A decrease was observed in the average number of cases recorded in REIV--TOXO: from 10 cases/year for 2015–2018 to 3 cases/year for 2019–2022.

### Pregnancy

During pregnancy, only 11% of mothers presented symptoms consistent with toxoplasmosis (fever, enlarged cervical lymph nodes, odynophagia, fatigue, headache, and/or myalgia). The trimester of maternal infection was determined in 49/54 mothers (90.7%) as follows: 5/54 (9.3%) in the first trimester, 11/54 (20.3%) in the second, and 33/54 (61.1%) in the third. In five cases, the time of maternal infection could not be established: three patients had no serology results (no prenatal screening) and the other two had only one determination, thus not allowing the moment of infection to be identified.

Amniocentesis was performed in a third (18/49; 36.7%) of the pregnant women, with the procedure done at a median gestational age of 29 weeks (IQR, 20–38) with PCR for *T. gondii* in AF and was positive in 12/18 cases (66.7%).

In 70% (37/53 –data were not available for one pregnancy–) of pregnancies, regular fetal ultrasound follow-up was performed each trimester. Ultrasound was normal in almost all cases (98%), except for one that showed intrauterine growth restriction. Fetal MRI was performed in three cases and was normal in all.

About a quarter of the mothers diagnosed during pregnancy (13/49, 26.5%) did not receive antiparasitic treatment during pregnancy. The reasons for failure to receive prenatal treatment were poor follow-up during pregnancy (2/13; 15.3%) and late diagnosis $\leq$ 7 days to delivery

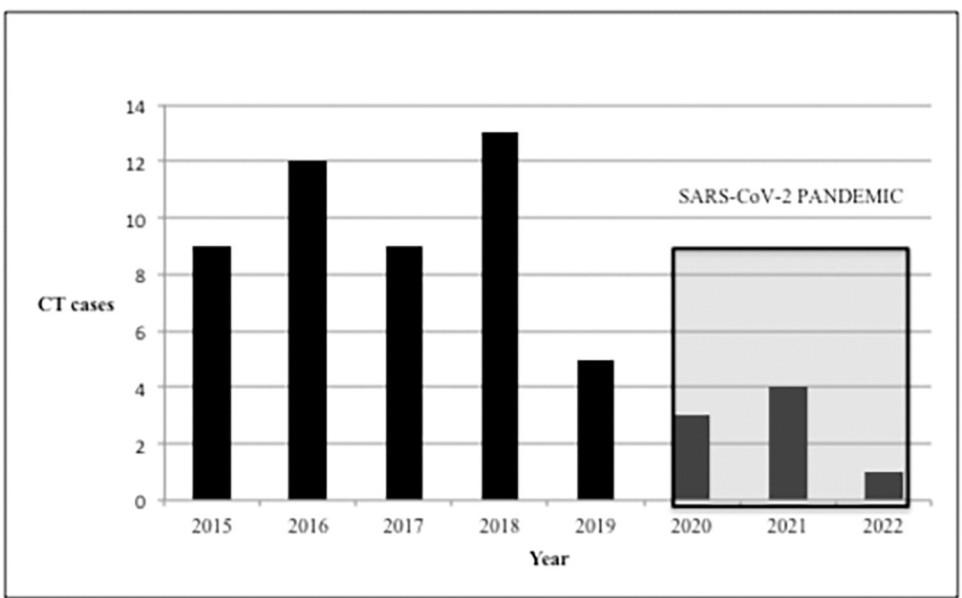

**Fig 1. Trends in CT cases detected in Spain.** The REIV-TOXO registry. *CT = congenital toxoplasmosis; REIV-TOXO = Spanish Research Network of Congenital Toxoplasmosis; SARS-CoV-2 = severe acute respiratory syndrome coronavirus 2.*

(6/13; 46.1%). No specific reason was reported for the remaining cases (5/13; 38.4%). In eight pregnancies, the duration of antiparasitic treatment was between 7 and 27 days (6 SPI and 2 PSA). Early discontinuation of prenatal treatment was explained by the end of pregnancy in all of these cases. Among those who were treated for $\geq 28$ days, 18/28 (64.2%) received SPI, 6/28 (21.4%) received PSA, and the remaining four (14.2%) received both treatments sequentially. The median number of treatment days was 66 for SPI (IQR, 6–198) and 51 for PSA (IQR, 19–91).

A positive PCR result for *T. gondii* in AF (12/18) during amniocentesis led to the start of PSA in 6/12 or SPI switch to PSA in 3/6 cases, with PSA being maintained until the end of the pregnancy in 8/9 cases. In two mothers, SPI was maintained until the end of pregnancy despite positive PCR for *T. gondii* in AF.

In 75% (27/36) of pregnancies, data were available on the time between diagnosis and treatment initiation, whether SPI or PSA, with the median being 11.3 days (IQR, 0–62). No adverse effects (AEs) were observed during treatment with SPI. One serious adverse event was reported during treatment with PSA, namely Stevens–Johnson syndrome, leading regimen discontinuation one month after it had been started.

## Infected newborns and infants

A total of 85.7% newborns were born at term, and 57% were male. Prenatal screening detected 92.8% (52/56) cases of CT in newborns subsequently confirmed at birth or during follow-up in the first year of life (Fig 2), while another four whose mothers had not undergone serological screening during pregnancy were diagnosed at birth (three cases) or during the first year of follow-up (one case) due to symptomatic CT.

Most newborns were asymptomatic at birth (35/56; 62.5%). Among the 21 newborns who were symptomatic at birth, 15 (26.7%) had CNS involvement, nine (16%) had ocular involvement, and four (7.1%) had generalized disease (Table 2). Notably, 28.5% of patients had two or

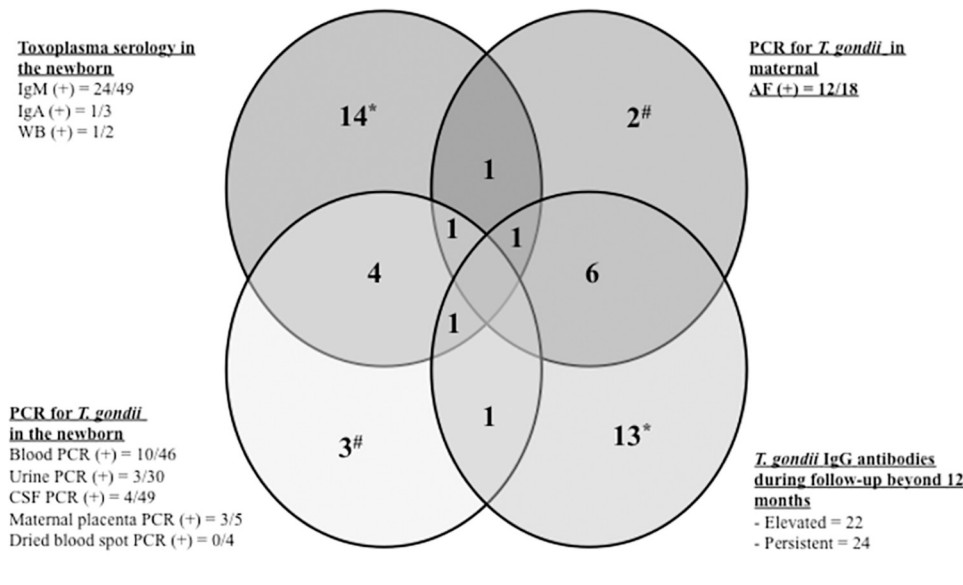

**Fig 2. Confirmatory microbiological tests for the diagnosis of toxoplasmosis performed on newborns.**
AF = amniotic fluid; WB = Western blot for T. gondii (mother-newborn).

more clinical forms: five patients had ocular and CNS involvement and one had generalized, ocular, and neurological disease. In newborns with neurological involvement, 8/15 (53.3%), the mother had undergone all three routine prenatal trimester ultrasounds with no abnormalities reported, although intracranial calcifications were later observed at birth in 5/8 of these infants.

A median of 364 days of treatment were completed by 47/56 (84%) of newborns (IQR, 11–494): 44/47 completed treatment with PSA and 3/47 had AEs requiring treatment to be temporarily switched to clindamycin alone (one case) or SPI (two cases). Among the nine patients who did not complete treatment, three were diagnosed after 12 months of age, four had treatment discontinued prematurely due to AEs, and two had poor adherence to treatment.

Globally, half the patients experienced AEs during treatment (27/53; 50.9%–data were not available for three children–). The most common AE was neutropenia (23/53; 43.3%); which required granulocyte colony-stimulating factor (G-CSF) treatment in four patients. Other AEs reported were crystalluria (one case), hepatotoxicity (one case), thrombocytosis (two cases), aplastic anemia (one case), hives (one case), and Stevens–Johnson syndrome (one case).

During a median follow-up of 24 months (IQR, 3–72), new infection-related complications (IRC) were reported in 8/56 (14.2%) of the children; 50% of these eight children were asymptomatic at birth and 75% did not receive treatment during pregnancy (Fig 3). Five developed retinochoroiditis resulting in loss of visual acuity (first episode of retinochoroiditis in 2/5; new foci of known retinochoroiditis in 3/5). Additionally, 4/56 (7.1%) showed new neurological involvement, including psychomotor delay (3/4), microcephaly (1/4), and spasticity (1/4); and 1/56 experienced new-onset neurosensorial hearing loss. Asymptomatic children at birth were less likely to develop new IRC during follow-up than symptomatic children (11.4% *vs.* 19%), although the differences were not significant (p = 0.43). No deaths were recorded in our cohort during the two-year follow-up period.

**Table 2. Clinical manifestations at birth of newborns with CT.**

| Ocular CT<br>n = 9 (16%) | CT with CNS involvement<br>n = 15 (26.7%) | CT with generalized disease<br>n = 4 (7.1%) |
|---|---|---|
| Retinochoroiditis<br>  Unilateral: n = 4 (44.4%)<br>  Bilateral: n = 3 (33.3%)<br>Macular involvement<br>n = 2 (28.5%) | Intracranial calcifications<br>n = 7 (46.6%) | Jaundice<br>n = 3 (75%) |
| Cataracts<br>n = 1 (11.1%) | CNS lesions<br>n = 7 (46.6%) | Enlarged spleen<br>n = 3 (75%) |
| Unilateral retinal hemorrhage<br>n = 1 (11.1%) | Ventriculomegaly<br>n = 1 (6.6%) | Hepatomegaly<br>n = 1 (25%) |
|  | Hydrocephalus<br>n = 1 (6.6%) |  |
|  | Microcephaly<br>n = 1 (6.6%) |  |

CNS = central nervous system; CT = congenital toxoplasmosis.

## Role of prenatal treatment

There was a significant association between receiving prenatal treatment (SPI and/or PSA) ≥ 28 days and the absence of symptoms at birth (p = 0.03), with newborns whose mothers did not receive prenatal treatment having a 4.1-fold risk of symptomatic CT at birth (Table 3). This effect remained significant regardless of the trimester of maternal infection. A statistically significant association (p = 0.01) was also observed between prenatal treatment lasting ≥ 28 days and the absence of neurological involvement at birth, with a 7.5-fold risk of developing neurological symptoms in newborns whose mothers did not receive antiparasitic treatment.

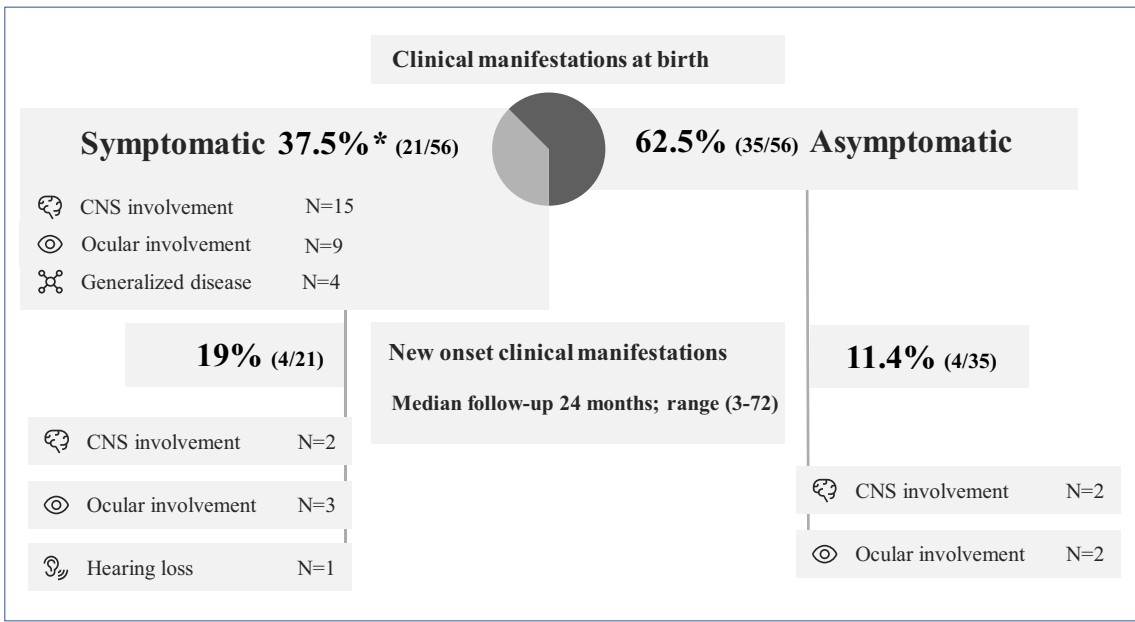

**Fig 3. Clinical manifestations at birth and new infection-related complications during follow-up.** *Five children had ocular and CNS involvement, and one child had all of the manifestations. *CNS = central nervous system.*

**Table 3. Association between prenatal treatment and newborn characteristics.**

| | Prenatal treatment | | | | Univariate analysis | | Multivariate analysis | |
|---|---|---|---|---|---|---|---|---|
| | No (n = 19) | | Yes (n = 28) | | | | | |
| | n | % | n | % | p | OR (95% CI) | p | OR (95% CI) |
| Confirmed maternal infection | | | | | | | | |
| 1st trimester | 3/15 | 15.78 | 2/27 | 7.14 | | | | |
| 2nd trimester | 1/15 | 5.26 | 9/27 | 33.33 | | | | |
| 3rd trimester | 11/15 | 57.89 | 16/27 | 57.14 | 0.17 | | | |
| Prematurity | 5/19 | 26.31 | 1/28 | 3.57 | 0.05 | 0.10 (0.01–1.03) | | |
| Symptomatic CT (all) | 11/19 | 57.89 | 7/28 | 25.00 | **0.03** | **4.12 (1.14–14.89)** | 0.08 | 3.64 (0.83–16.04) |
| Ocular CT | 5/19 | 26.32 | 4/28 | 14.28 | 0.31 | | 0.34 | |
| CNS C | 9/19 | 47.37 | 3/28 | 10.71 | **0.01** | **7.50 (1.60–34.97)** | 0.08 | 4.5 (0.80–25.11) |
| Generalized CT | 3/19 | 15.78 | 1/28 | 3.57 | 0.18 | | 0.86 | |
| New IRC (all) | 6/19 | 31.58 | 2/28 | 7.14 | **0.04** | **6.00 (1.01–35.61)** | **0.03** | **14.11 (1.31–152.06)** |
| New ocular IRC | 4/19 | 21.05 | 1/28 | 3.57 | 0.09 | 7.20 (0.69–74.99) | 0.06 | 12.71 (0.81–198.65) |
| New CNS IRC | 3/19 | 15.79 | 1/28 | 3.57 | 0.18 | | 0.12 | |

CI = confidence interval; CNS = central nervous system; CT = congenital toxoplasmosis; IRC = infection-related complications; OR = odds ratio.

Surprisingly, there were no statistically significant differences in clinical manifestations at birth between newborns whose mothers were treated with SPI and those treated with PSA (p = 0.79).

Significant differences (p = 0.04) were also observed in new IRC occurring during follow-up, indicating that children born to untreated mothers were six times more likely to experience new IRC during follow-up, a finding that remained significant regardless of the trimester of maternal infection (Table 3).

## Discussion

The national CT cohort of REIV-TOXO provides updated information on the status of CT in Spain, and the data may be representative of the current situation in Southern Europe. In our series, prenatal screening each trimester detected toxoplasmosis during pregnancy in most cases. This made it possible to offer treatment to the pregnant mother, confirm most CT cases at birth or during the first year of life, and provide postnatal treatment with few treatment-related AEs. Infants treated during the gestational period had fewer symptoms at birth (four-fold lower and fewer IRC during follow-up (six-fold lower risk). These results confirm the importance of prenatal treatment and underscore the need for prenatal screening programs for *T. gondii*.

Several factors may explain the decrease in the number of cases recorded in the REIV--TOXO database since 2018: the possible effect of the SARS-CoV-2 pandemic on case diagnosis and reporting as happened in the case of other congenital infections [33], a potential reduction in *T. gondii* seroprevalence among pregnant women in Spain due to lower circulation of the parasite and a lower risk of contracting the infection; the decrease in birth rates; and the gradual withdrawal of prenatal screening for *T. gondii* in some regions of Spain, the latter likely being the most influential factor. Differences in the reporting rate from each autonomous community are mainly related to the current availability of pregnancy screening programs in their portfolio of services, as discussed in depth in a previous article from our group [28].

In our cohort, clinical symptoms in pregnant women and ultrasound abnormalities suggestive of CT showed a low sensitivity for identifying all cases, as described in previous studies [4,22,34–36]. Thus, if screening had not been performed, most children with CT would have been missed and left untreated.

Although published studies about the efficacy of prenatal treatment in preventing maternal-fetal transmission and its impact on birth outcomes between 1999 and 2007 fueled controversy on this issue, they had major methodological biases [37–42]. More recent studies have shown that prenatal treatment appears to be a decisive factor in reducing transplacental transmission and the appearance of any IRC in concordance with our findings [12,43–50]. In our opinion, although prenatal treatment has been shown to be beneficial with regard to the most common complication in our cohort, i.e. neurological manifestations, the relatively low number of newborns with nonneurological forms of the disease may have made it difficult to show this potential benefit in these specific situations, as also seen for prematurity which remains marginally significant (p = 0.05). The decision to exclude the "uncertain prenatal treatment group" from the analysis was based on the lack of evidence on the minimum duration of prenatal treatment that would affect fetal infection. While less than 7 days seems a good theoretical cut-off for defining insufficient treatment duration, we selected 28 days of treatment as the minimum acceptable length of treatment to evaluate the impact of prenatal treatment. Interestingly, the beneficial effects of prenatal treatment remained unchanged when the analysis included prenatal treatment of more than 7 days in the treatment group (**S1 Appendix**). According to these findings, prenatal treatment should be started regardless of the proximity of the delivery date, despite uncertainties about the benefit of short-term treatment.

The only published clinical trial found that PSA showed greater efficacy than SPI in preventing maternal-fetal transmission, although the differences were not statistically significant, probably due to a lack of power [51] and, at present, PSA is considered the "gold standard" for the treatment of gestational toxoplasmosis. Surprisingly, our study showed no differences in efficacy between SPI–an antibiotic with lower in vitro activity against *T. gondii* and poor transfer to the fetus, but with considerable capacity of concentration in placental tissue [43]–and PSA in preventing the appearance of symptoms at birth and subsequent IRC. Although these findings may be due to the relatively small sample size of our study, a recent meta-analysis supports the role of SPI in reducing maternal-fetal transmission and could help limit IRC in infected fetuses regardless of serious illness [52,53], an effect also observed in our cohort. However, when fetal infection is confirmed after week 14 or when maternal infections are detected in the third trimester and no amniocentesis is performed, starting with PSA or switching from SPI to PSA remains the current recomendation [19].

Several studies have observed a "window of opportunity" for treatment, which has been shown to be more effective if administered in the first three weeks after maternal infection [42–51]. In our series, the median time between the diagnosis of gestational toxoplasmosis and the start of treatment was 11 days, which could enhance the beneficial effect of gestational treatment, even if the actual time of infection in the pregnant mother is unknown due to the limitations of trimester-based screening. If serological screening is performed once each trimester, there is greater uncertainty about the time of maternal infection because it could have happened some weeks before diagnosis, making monthly screening the most effective approach. In France, the introduction of routine monthly gestational screening and PCR for *T. gondii* in AF has made it easier to start early prenatal treatment and has significantly lowered subsequent symptomatic cases, neurological IRC, and death [45]. However, although monthly screening is the most accurate, it is associated with higher costs and more medical visits during pregnancy. The novel introduction of point-of-care tests for rapid CT diagnosis could facilitate the development of well-accepted monthly gestational screening programs, even in countries with a lower incidence of CT [54,55]. In the absence of a monthly screening approach, screening once per trimester has also been shown to be a useful and necessary tool, as confirmed by our study.

Both SPI and PSA can cause AEs in pregnant women and their children, with reversible neutropenia being the most commonly reported AE in children, similar to other published series [56–60]. Two cases of Stevens–Johnson syndrome (one mother and one newborn) were also observed but responded well to treatment interruption and supportive therapy. This serious AE is rare and has been reported in 0.1% of patients treated with pyrimethamine-based regimens and sulfonamides [61]. To detect AEs in children treated with PSA for CT, hematological and clinical follow-up is currently recommended.

In our study, the low sensitivity (66.7%) of PCR for *T. gondii* in AF is in contrast with the high sensitivity (87–92%) and high specificity (99–100%) previously reported for this technique [4,5,19,20,22,62]. The fact that 5/6 false negative results correspond to gestational infections in the first and second trimester, when the test has been reported to have a lower sensitivity, may partially explain our results [5,20]. In our cohort, only 66% of pregnant women with positive amniocentesis were treated with PSA until the end of their pregnancy. Future improvement in the management of gestational toxoplasmosis should address the poor performance of amniocentesis and the lack of adequate prenatal treatment observed in a quarter of the cases studied. This technique should be routinely offered as a diagnostic tool within the diagnosis–therapeutic algorithm for toxoplasmosis, given the safety of the procedure and the minimal risk of complications [63,64]. Nevertheless, if PCR is not performed for medical reasons or as decided by the pregnant women, the start of treatment is recommended when a first infection is clearly suspected in the mother, even if congenital transmission cannot be confirmed.

The absence of mortality in our series is consistent with the mortality rate observed in Europe, where it is less than 1% of cases [37,46]. Serious forms of the illness are seen more often in countries without prenatal screening [53,65]. In our cohort, the most common symptomatic form at birth was neurological, unlike other studies published in Europe where ocular forms were predominant at birth [42,45,66,67].

Multidisciplinary follow-up of the patient until adulthood is crucial to detecting and treating any future IRC. In our series, the appearance of new episodes of retinochoroiditis was the most common complication during follow-up, accounting for 40% of all first episodes in previously asymptomatic patients. In Europe, the most common IRC is retinochoroiditis, with a recurrence rate of 18% to 31% in patients receiving postnatal treatment [66,68–72], a slightly higher incidence than that observed in our cohort. Interestingly, one patient in our study presented with neurosensorial hearing loss during follow-up, a complication also reported by other studies [5,39,73], highlighting the importance of long-term follow-up of these patients.

Our study had several limitations. First, newborns and infants are included in the REIV-TOXO cohort on a voluntary basis. Thus, it does not offer a full picture of CT in Spain as confirmed by the 16 cases never reported to the REIV-TOXO cohort (despite our efforts to contact collaborators to encourage them to enter the data), and other cases reported in databases of the National Centers of Microbiology and Epidemiology of Spain. Moreover, symptomatic CT during early childhood who develop IRC later in adolescence or adulthood may also be missed. Additionally, REIV-TOXO did not collect information on abortions or miscarriages due to *T. gondii*. In view of the current status of toxoplasmosis screening during pregnancy in Spain, our cohort may only represent the tip of the iceberg of the problem. Despite all this, our registry represents the highest number of cases registered in Spain, much higher than those reported by the public health authorities. Second, the characteristics of our series did not allow a multivariate analysis to investigate any possible association between prenatal treatment and outcomes adjusted for the gap between maternal infection and prenatal treatment ("window of opportunity"), and it also failed to assess the appearance of new IRC during follow-up by controlling for the effect of postnatal treatment. Third, most of our patients had no records

of coinfection with other congenital infections, in particular, coinfection with cytomegalovirus, which could involve more serious neurologic manifestations [46]. Another limitation is that brain MRI was performed in only 19/54 of newborns, which could underestimate CNS involvement at birth. Nevertheless, in most of our patients (10/15) with CNS involvement at birth, imaging abnormalities were seen on ultrasound, an examination performed in 98% of cases. Last, mean follow-up was only two years and, therefore, our results underestimate the true extent of long-term CT morbidity.

In conclusion, because cases detected by prenatal screening and treatment with SPI and/or PSA had fewer complications at birth and during follow-up, we strongly recommend implementing universal screening in Spain and in countries with similar epidemiological data where the REIV-TOXO cohort could expand. Long-term follow-up of our cohort will provide further information on late complications and on the possible effects of prenatal and postnatal treatment.

## Supporting information

**S1 Appendix. Association between prenatal treatment and newborn characteristics including the "uncertain prenatal treatment group" (7–27 days of treatment) in the analysis within the prenatal treatment group.**
(DOCX)

**S1 Acknowledgments. Members of the Spanish REIV-TOXO group.**
(DOCX)

## Acknowledgments

We would like to thank all the patients and families for their participation in REIV-TOXO, in particular the Bescos-Manau family for its unconditional support to the project, as well as all the investigators in the network, Ana Vázquez for statistical support, and Helen Casas and Laura Casas for linguistic support.

## Author Contributions

**Conceptualization:** Borja Guarch-Ibáñez, Clara Carreras-Abad, Marie Antoinette Frick, Daniel Blázquez-Gamero, Fernando Baquero-Artigao, Pere Soler-Palacin.

**Data curation:** Borja Guarch-Ibáñez, Clara Carreras-Abad, Marie Antoinette Frick, Pere Soler-Palacin.

**Formal analysis:** Borja Guarch-Ibáñez, Clara Carreras-Abad.

**Investigation:** Borja Guarch-Ibáñez, Clara Carreras-Abad, Marie Antoinette Frick, Pere Soler-Palacin.

**Methodology:** Borja Guarch-Ibáñez, Clara Carreras-Abad, Marie Antoinette Frick, Pere Soler-Palacin.

**Project administration:** Borja Guarch-Ibáñez, Clara Carreras-Abad, Marie Antoinette Frick, Daniel Blázquez-Gamero, Fernando Baquero-Artigao, Isabel Fuentes, Pere Soler-Palacin.

**Resources:** Borja Guarch-Ibáñez, Clara Carreras-Abad, Marie Antoinette Frick.

**Software:** Borja Guarch-Ibáñez.

**Supervision:** Clara Carreras-Abad, Daniel Blázquez-Gamero, Fernando Baquero-Artigao, Isabel Fuentes, Pere Soler-Palacin.

**Validation:** Borja Guarch-Ibáñez, Clara Carreras-Abad, Marie Antoinette Frick, Daniel Blázquez-Gamero, Fernando Baquero-Artigao, Isabel Fuentes, Pere Soler-Palacin.

**Visualization:** Borja Guarch-Ibáñez, Clara Carreras-Abad, Marie Antoinette Frick, Isabel Fuentes, Pere Soler-Palacin.

**Writing – original draft:** Borja Guarch-Ibáñez, Clara Carreras-Abad, Marie Antoinette Frick, Daniel Blázquez-Gamero, Fernando Baquero-Artigao, Isabel Fuentes, Pere Soler-Palacin.

**Writing – review & editing:** Borja Guarch-Ibáñez, Clara Carreras-Abad, Marie Antoinette Frick, Daniel Blázquez-Gamero, Fernando Baquero-Artigao, Isabel Fuentes, Pere Soler-Palacin.

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
