## [Decision Letter · Decision Letter 0]

6 Jun 2024

Dear Mr. Guarch-Ibañez,

Thank you very much for submitting your manuscript "REIV-TOXO Project: Results from the Spanish cohort of congenital toxoplasmosis (2015-2022)" for consideration at PLOS Neglected Tropical Diseases. As with all papers reviewed by the journal, your manuscript was reviewed by members of the editorial board and by several independent reviewers. In light of the reviews (below this email), we would like to invite the resubmission of a significantly-revised version that takes into account the reviewers' comments. 

We cannot make any decision about publication until we have seen the revised manuscript and your response to the reviewers' comments. Your revised manuscript is also likely to be sent to reviewers for further evaluation.

Sincerely,

Claudia Ida Brodskyn

Section Editor

Claudia Brodskyn

Section Editor

Reviewer's Responses to Questions

**Key Review Criteria Required for Acceptance?**

**Methods**

-Are the objectives of the study clearly articulated with a clear testable hypothesis stated?

-Is the study design appropriate to address the stated objectives?

-Is the population clearly described and appropriate for the hypothesis being tested?

-Is the sample size sufficient to ensure adequate power to address the hypothesis being tested?

-Were correct statistical analysis used to support conclusions?

-Are there concerns about ethical or regulatory requirements being met?

Reviewer #1: � The exhaustivity of this cohort must be doubted given that 22% of diagnosed cases were not included. At least, regional data about the parasite strain types in human disease and the prevalence of seropositivity should be provided along with the number of life births per year during the same period (also to be added to figure 1). 

That 16 children with confirmed CT could not be included in the registry limits representativity. Attempts should be made to get records of these children, at least a root data set to demonstrate that these were similar in the severity of affection to the included patients.

What seems worrisome to me is the duration of treatment during pregnancy. Once the decision in favour of treatment is made, any established guideline clearly states the need for maternal treatment until birth, while in this series, mothers were categorized as having received treatment if they had received it for at least 7 days (line 157). Ther is neither clinical nor experimental evidence that a treatment of such a short duration is of any impact to the fetus, and namely not for preventing infection. I have no clue, how the authors want to deal with this point. Personally, I would choose a minimal treatment duration of 4 weeks for tailoring mothers to any group, or provide 3 groups (for the price of a further reduction of statistical power) with one group with treatment up to 7 days (no treatment group), one with longer intercurrent treatment, and finally one with treatment from diagnosis until delivery, which represents the recommended standard, once a decision to treat has been made. Generally, a median time of 37 days under treatment is away from treatment standards and a very critical point. This might well explain why the portion of children with cerebral affections was remarkably high compared to other cohorts (line348-52). Moreover, possibly, the number of children with ocular manifestation of their CT is relatively low so that the experience of the ophthalmologists in examining newborns with CT was granted. The merely unpigmented lesions at birth are definitely easily missed, even in experienced hands. . 

A delay of 11 days between diagnosis and treatment is short, but not meaningful, if the time of contamination cannot be estimated. Therefore, more detailed information about the gestational situation at contamination might be provided if possible. 

Since the topic of this paper is to assess the impact of prenatal screening and treatment, the case diagnosed during the 1st year of life should be excluded from the analysis not having received either of both (line 220). 

Random screening and systematic screening data must not be pooled. I would recommend to differentiate pregnancies where a systematic screening was performed (at least 1x per trimester) and those without a systematic screening to make sure the interval between infection and its detection can be estimated; please add systematic screening to reporting as an independent confounder with systematic screening possibly being more relevant than unsystematic reporting (line 94). 

The application of a logistic regression model is n not meaningful as long as the time gap between infection and treatment initiation and the duration of treatment are not included. Any results regarding regimen and outcomes are therefore useless and need to be removed.

Reviewer #2: I have doubts about the definition of the population of infected newborns

The sample size is sufficient

No ethical concerns

There are no concerns regarding compliance with ethical or regulatory requirements

Reviewer #3: the study design is appropriate to address the stated objectives

the population clearly described and appropriate for the hypothesis being tested

the sample size is sufficient to ensure adequate power to address the hypothesis being tested

statistical analysis is correct to make any conclusion

**Results**

-Does the analysis presented match the analysis plan?

-Are the results clearly and completely presented?

-Are the figures (Tables, Images) of sufficient quality for clarity?

Reviewer #1: � Given that only 54 pregnancies were registered from 122 hospitals over a period of 7 years implies that either a significant number of pregnancy-associated cases of toxoplasmosis have been missed or that the problem is on the decline: a threefold reduction in the number of cases per year is not in favour of a significant health concern due to CT in Spain. On the other hand, each single case with organ damage that can be prevented is a life changer for the affected families. As the authors mention, a systematic screening was not performed, which means that their cohort may be a selection of more severely affected pregnancies or at least an uncertainty of the obstetricians triggering maternal blood testing. This is indicated by a higher than expected prevalence of symptomatic children (table 3). 

As long as not proven different, the data may be representative of a European cohort, which needs to be stated in line 85. 

Generally, the parasites have a tropism to the neuro-epithelium, which means they primarily affect the retina and only secondarily spread to the choroid. Affected patients thus pathophysiologically present a retinochoroiditis, but not a chorioretinitis. Please change the wording throughout. 

Cost coverage may be a relevant factor differentiating systematic from non-systematic screening. Since social situation is an important influence factor, as mentioned by the authors, this point deserves acknowledgement. Mothers receiving full cost coverage should be compared to those paying out of the pocket regarding time between presumed infection and its diagnosis and treatment. 

Was the 3-fold reduction in cases recorded between 2019-22 linked to changes in the screening strategy or cost coverage (lines 174-8)? As mentioned, the number of life births needs to be compared during the 2 periods to allow an interpretation of these findings. 

That “Newborns born from mothers treated prenatally had a reduced risk of CT clinical features at birth and a lower risk of further complications during follow-up” is supported by the data, while I do not see a four-fold and an eight-fold lower risk after talking diagnostic and therapeutic differences into account. A portion of 32% of IRC over 24 months in untreated patients, again, is significantly higher than expected from other European series. One reason for this might be that the portion of immigrants from geographical areas with a higher virulence is higher than in not spanish-speaking countries (see also lines 91-2). Are there data around to address this impact factor? The sentence (lines 286-7) is not supported by their data and must be removed.

What could an obstetrician drive to undertake amniocentesis at week 38 when the game is over for influencing pregnancy. Please report the standards on which amniocentesis is usually done (line 194). 

The term “ambispective” is not generally used and not understandable in the context of this heterogenous cohort.

Reviewer #2: Table 3 should be redone. 

I have doubts about the multivariate analysis and the definitions of infected newborn used

Reviewer #3: Results are clearly presented

**Conclusions**

-Are the conclusions supported by the data presented?

-Are the limitations of analysis clearly described?

-Do the authors discuss how these data can be helpful to advance our understanding of the topic under study?

-Is public health relevance addressed?

Reviewer #1: � It is not a failure, but an active decision not to treat (line 200).

Repetitions from results in text and tables need to be removed, Given the low number of observations, 1-2 tables might do, table 3 should be removed given the lack of strength and standardisation of treatment, as outlined above. 

The impact of 1 year of postnatal treatment onto outcomes deserves to be discussed, given the relatively high number of cases developing secondary complications. Please add this point in the discussion (lines 310 ff).

Reviewer #2: The conclusions are sufficiently supported by the data presented

-Multivariate analysis should be checked

The authors sufficiently discuss how these data can be useful to improve knowledge on Toxoplasmosis and to improve the prevention strategy

Reviewer #3: The conclusions support the data presented.

Some suggestions/comments are introduced in the letter to the authors.

**Editorial and Data Presentation Modifications?**

Reviewer #1: (No Response)

Reviewer #2: (No Response)

Reviewer #3: “Minor Revision”

**Summary and General Comments**

Reviewer #1: In the absence of solid prospective evidence for screening and intrauterine infection prophylaxis, and, if too late, therapy of children with CT, a systematic collection of retrospective data, as done in the “REIV-TOXO Project: Results from the Spanish cohort of congenital toxoplasmosis (2015-2022)” is urgently needed. In so far, I congratulate the authors to their activities, which are highly appreciated. Nevertheless, the strength of each single of these studies is not sufficiently strong to convince political opinion makers regarding its cost-effectiveness, while efficacy is principally accepted. As such, the REIV-TOXO Project adds to existing data, while a significant inherent limitation of this project is its observational character, since neither diagnostic not therapeutic standards have been applied explaining the heterogenicity of their data as visible by the broad standard deviations. This should have strong impact on the interpretation of their data and induce a critical discussion, which I have not seen. 

I fully agree that a systematic screening, and be it on a minimal scale of once per trimester, is justified, if this triggers a consequent maternal treatment until delivery which was not the case in this series. This limits the strength of data and possible conclusions based thereon. From a public health perspective, cost-effectiveness is to be shown. What are the costs of preventive measures and prophylactic therapy compared to the costs induced by infected individuals. Public health physicians have not necessarily an idea what CT could mean for the affected individual and its family. And their primary perspective is unfortunately a reduction of health care costs.

Reviewer #2: PLOS Neglected Tropical Diseases 

REIV-TOXO Project: Results from the Spanish cohort of congenital toxoplasmosis (2015-2022) 

It is known that toxoplasmosis contracted during pregnancy can cause congenital infection of the newborn and serious outcomes. The burden of this disease in the EU/EEA cannot be assessed due to large differences in national surveillance systems, screening programs and follow-up procedures of pregnant women. In addition to implementing national surveillance strategies, prevention of congenital toxoplasmosis should be implemented. Pregnant women should always receive information on the risks of exposure to T gondii and on the preventive measures to be taken. Serological screening during pregnancy, given that there is an effective therapy, should be implemented. The manuscripts that provide data are very relevant. 

Author summary, page 5 lines from 67 to 70

Authors: We also focused on the effect of prenatal treatment on clinical outcomes at birth and during follow-up, since its effectiveness is still controversial and may raise several concerns. One of these issues is the actual benefit of serological screening during pregnancy as part of a public health strategy to mitigate the disease. 

Reviewer: I believe that no one can contradict the usefulness of carrying out prenatal screening for toxoplasmosis during the pregnancy, given that there is a therapy which, regardless of the data collected in Spain, is proven to prevent the onset of feto-neonatal damage or in any case limit its severity. 

I would say to rephrase the sentence.

Page 5 line 71

Authors: Although the usefulness of prenatal screening is still under debate, …..

Reviewer: there is a therapy which is proven to prevent the onset of damage in fetuses and in neonates or in any case limit its severity. Please rephrase the sentence.

Authors: prenatal screening is still under debate, it remains the only tool available to diagnose all children …….

Reviewer: Diagnosis of infection in pregnancy is not a tool for diagnosing infection in children.... please rephrase.

Page 6 line 96

Authors: At present, prenatal serological screening is the only diagnostic tool available to identify all …….

Reviewer: as mentioned above the sentence is improper. Prenatal screening identifies infected mothers. Please rewrite the sentence.

Page 6 line 98

Reviewer: Please add some other informations “Toxoplasmosis in pregnancy is generally asymptomatic and the diagnosis is based only on serological tests. Some Countries such as France Austria Italy promote regular serological screening of pregnant women. Other countries have no prenatal surveillance program at all. The reasons for these differences are multiple and debatable. These include the effectiveness of prenatal treatment and the cost-effectiveness of such a program. Nevertheless, a significant body of data has demonstrated that rapid onset of treatment after maternal infection reduces the risk and severity of fetal infection.

Page 8 line 125

Authors: The inclusion criteria for a confirmed CT case in the REIV-TOXO registry …..

Reviewer: The precise wording would be: 

"Newborns who showed: positivity of ....." were defined as newborns affected by congenital Toxoplasmosis, symptomatic or non-symptomatic.

Newborns defined as having Congenital Toxolasmosis were reported through the registry....

It would be best to identify a separate paragraph within the methods section, which contains definitions and then state who was reported through the registry.

Page 9 lines 156-157

Authors: The same model was also used to compare the two regimens received: SPI and PSA, with SPI and/or PSA treatment defined as treatment received for at least seven days.

Reviewer: It would also be interesting to know the effect on the infectious status of the newborn at the gestational age at which antiprotozoal therapy began, because Toxoplasma crosses the placenta more in the last trimester of gestation.

Page 11 lines 193-195

Authors: Amniocentesis was performed in a third (18/49; 36.7%) of the pregnant women, with the

procedure done at a median gestational age of 29 weeks (IQR, 20–38) with PCR for T. gondii in AF

being positive in 12/18 cases (66.7%) …

Reviewer: Why amniocentesis so late? Why such a low number of cases during the pandemic? Please, discuss these particularities.

Page 16 lines from 266 to 268

Authors: A statistically significant association (p = 0.01) was also observed between prenatal treatment and the absence of neurological involvement at birth, with a 4.65-fold risk of developing neurological symptoms in newborns whose mothers did not receive antiparasitic treatment (OR, 4.65; 95% CI, 1.28–16.78). 

Reviewer: It would be better to express the association between the absence of maternal therapy and feto-neonatal neurological damage. How can we explain the lack of significance of the multivariate association in table 3?

Is it necessary to recheck the case definition? In Europe all the reporting Member States used the EU case definition from 2008, 2012, or 2018 (the case definition remained the same), except two which used other (not specified) case definitions.

In summary

the manuscript is very important because it provides neonatal data on a topic on which there is fragmentary epidemiological knowledge. The data in the table should be rechecked for the association between the dependent variables (symptoms, neonatal neurological damage, ocular damage) and the risk factor "absence of treatment during pregnancy".

To be re-evaluated after corrections.

Reviewer #3: The paper titled "Results from the Spanish cohort of congenital toxoplasmosis (2015-2022)" by Guarch-Ibanez et al. highlights the importance of maternal screening for T. gondii in Spain and treatment for the management of congenital toxoplasmosis. The conclusions drawn from this paper support the implementation of a mandatory prenatal screening program followed by treatment for women who test positive to reduce the impact of the disease on newborns and during follow-up. Here, we applaud the authors for citing on many occasions France, a country with historically successful management of congenital toxoplasmosis, and a similar incidence rate of CT in Spain, which suggests that prenatal screening is cost-effective compared to neonatal screening. This approach also empowers pregnant women to educate themselves on how to reduce the risk of infection and decrease the incidence of toxoplasmosis in the country. However, the data from the Spanish cohort lacks the monthly serological screening of pregnant women, which creates uncertainty about the timing of maternal infection before diagnosis and the potential delay in treatment following seroconversion. Despite this, the paper is well-written and interesting to read. I recommend that the authors emphasize these key points to enhance the presentation of this manuscript.

1. The current title of the manuscript does not accurately represent its content and should be changed. 

2. The introduction and conclusion of the manuscript need further development. Please provide more details on why certain European countries (such as the UK, Denmark, and Sweden) and specific regions in Spain have rejected prenatal screening. Some argue that this decision is due to the clinical harms associated with false-positive diagnoses, the low and uncertain benefits of treatment, the harms associated with treatment, and the high costs of prenatal screening. 

3. Are there any studies from the Spanish cohort on the genetics of the parasite that could explain the severity observed in certain newborns, such as the case of a newborn with hydrocephalus? 

4. Technical information is necessary to understand how the diagnosis and screenings of pregnant women were reported in this cohort. Was the avidity test used to confirm those who are seropositive and not required for the screening? Which confirmation test for IgG/IgM was used? Do all these hospitals use the same confirmatory test? Is there any consensus on the choice of this test?

5. Please provide information on whether pregnant women who tested positive for toxoplasmosis had any adverse pregnancy outcomes, such as miscarriages or preterm births. Did you observe any differences in outcomes between toxoplasmosis-positive and negative women or between those who received treatment and those who did not? 

6. The authors should provide evidence to support their claim about the impact of SARS-CoV-2 on the decrease in reported cases of T. gondii infection in mothers in Spain since 2018 and whether similar trends are seen in other congenital diseases. Please include any relevant references from the literature.

7. It would be helpful to enhance the discussion by emphasizing the importance of new diagnostic tests (e.g. rapid tests) that can be incorporated into prenatal screening to alleviate the challenges of monthly screening in a prospective study of this cohort.

PLOS authors have the option to publish the peer review history of their article (what does this mean?). If published, this will include your full peer review and any attached files.

Reviewer #1: Yes: Justus G. Garweg

Reviewer #2: No

Reviewer #3: Yes: KAMAL EL BISSATI
---

## [Decision Letter · Decision Letter 1]

20 Sep 2024

Dear Mr. Guarch-Ibañez,

Thank you very much for submitting your manuscript "REIV-TOXO Project: Results from a Spanish cohort of congenital toxoplasmosis (2015-2022). The beneficial effects of prenatal treatment on clinical outcomes of infected newborns" for consideration at PLOS Neglected Tropical Diseases. As with all papers reviewed by the journal, your manuscript was reviewed by members of the editorial board and by several independent reviewers. The reviewers appreciated the attention to an important topic. Based on the reviews, we are likely to accept this manuscript for publication, providing that you modify the manuscript according to the review recommendations. 

Sincerely,

Claudia Ida Brodskyn

Section Editor

Claudia Brodskyn

Section Editor

Reviewer's Responses to Questions

**Key Review Criteria Required for Acceptance?**

**Methods**

-Are the objectives of the study clearly articulated with a clear testable hypothesis stated?

-Is the study design appropriate to address the stated objectives?

-Is the population clearly described and appropriate for the hypothesis being tested?

-Is the sample size sufficient to ensure adequate power to address the hypothesis being tested?

-Were correct statistical analysis used to support conclusions?

-Are there concerns about ethical or regulatory requirements being met?

Reviewer #1: Methodological weakness needs acknowledgement in the manuscript. 

The exclusion of 16 children with confirmed CT from the registry limits representativity as the authors state. The explanation given to the reviewer should be added to the manuscript: “REIV-TOXO is a voluntary registry, where the investigators at each site are responsible for entering the data, limit the authors' access to all cases, and thus there are 16 cases that were never entered despite our efforts to contact the collaborators on repeated occasions. Additionally, without parental consent, we are unable to provide this information. Despite this, our registry represents the highest number of cases registered in Spain, much higher than those reported by the public health authorities.”

Reviewer #2: The authors have accepted the revoewers' suggestion and to my opinion the manuscript can be published as it appears now.

Reviewer #3: (No Response)

**Results**

-Does the analysis presented match the analysis plan?

-Are the results clearly and completely presented?

-Are the figures (Tables, Images) of sufficient quality for clarity?

Reviewer #1: The requested information regarding regional data about the parasite strain types (which type is prevalent in Spain, 1, type 2, recombinant strains) in human disease. What is the prevalence of seropositivity in pregnant women, and what was the number of recorded life births with CT per year during the same period? These points have remained unanswered and deserve to be added to the manuscript. Genotyping data of given patients were not requested.

Reviewer #2: (No Response)

Reviewer #3: After revision, the analysis of results looks clearer.

**Conclusions**

-Are the conclusions supported by the data presented?

-Are the limitations of analysis clearly described?

-Do the authors discuss how these data can be helpful to advance our understanding of the topic under study?

-Is public health relevance addressed?

Reviewer #1: no further changes needed

Reviewer #2: (No Response)

Reviewer #3: Yes. The data support the conclusions.

**Editorial and Data Presentation Modifications?**

Reviewer #1: mostly fine for me

Reviewer #2: (No Response)

Reviewer #3: (No Response)

**Summary and General Comments**

Reviewer #1: ok

Reviewer #2: (No Response)

Reviewer #3: The authors have extensively reviewed their manuscript and have carefully addressed the reviewers' comments. The manuscript reads much better. After revision, the overall quality of the manuscript has improved to a level acceptable for publication in PLOS Neglected Tropical Diseases.

PLOS authors have the option to publish the peer review history of their article (what does this mean?). If published, this will include your full peer review and any attached files.

Reviewer #1: No

Reviewer #2: No

Reviewer #3: Yes: KAMAL EL BISSATI

Figure Files:

Data Requirements:

Reproducibility:

References

---

## [Editor Report · Decision Letter 2]

9 Oct 2024

Dear Mr. Guarch-Ibañez,

We are pleased to inform you that your manuscript 'REIV-TOXO Project: Results from a Spanish cohort of congenital toxoplasmosis (2015-2022). The beneficial effects of prenatal treatment on clinical outcomes of infected newborns' has been provisionally accepted for publication in PLOS Neglected Tropical Diseases.

Best regards,

Claudia Ida Brodskyn

Section Editor

Claudia Brodskyn

Section Editor

---

## [Editor Report · Acceptance letter]

17 Oct 2024

Dear Mr. Guarch-Ibañez,

We are delighted to inform you that your manuscript, "REIV-TOXO Project: Results from a Spanish cohort of congenital toxoplasmosis (2015-2022). The beneficial effects of prenatal treatment on clinical outcomes of infected newborns," has been formally accepted for publication in PLOS Neglected Tropical Diseases.

Best regards,

Shaden Kamhawi

co-Editor-in-Chief

Paul Brindley

co-Editor-in-Chief
